# Risk Factors for Low Quality of Life among Women Using Different Types of Contraceptives in Saudi Arabia: A Questionnaire-Based Study

**DOI:** 10.3390/bs14090829

**Published:** 2024-09-17

**Authors:** Malak M. Alhakeem, Leena R. Baghdadi, Almaha H. Alshathri, Aljohara H. Alshathri, Arwa A. Alqahtani, Monerah H. Alshathri

**Affiliations:** 1Obstetrics and Gynecology Department, College of Medicine, King Saud University, Riyadh 11362, Saudi Arabia; malhakeem@ksu.edu.sa; 2Department of Family and Community Medicine, College of Medicine, King Saud University, Riyadh 11362, Saudi Arabia; 3College of Medicine, King Saud University, Riyadh 11362, Saudi Arabia; 439200306@student.ksu.edu.sa (A.H.A.); 439200305@student.ksu.edu.sa (A.H.A.); 439200282@student.ksu.edu.sa (A.A.A.); 4Health Administration, King Saud University, Riyadh 11362, Saudi Arabia; 442204007@student.ksu.edu.sa

**Keywords:** contraception, quality of life, women, oral pills, sexual health

## Abstract

This study aimed to assess and identify the risks for poor quality of life among female Saudi contraceptive users by administering an online questionnaire. The validity of the Arabic version of the Spanish Society of Contraception Quality of Life (SEC-QOL) questionnaire was assessed by incorporating the relevant items into an exploratory factor analysis and a subsequent confirmatory factor analysis. Internal consistency was assessed using Cronbach’s alpha coefficient. *p* < 0.05 was statistically significant. Questionnaires completed by 652 eligible women were analyzed. The most common contraception method was oral pills (51.5% of respondents). Analysis of the internal consistency of the questionnaire revealed that overall reliability was >0.7, which is considered adequate (Cronbach’s alpha = 0.845). Analysis of risk factors associated with higher overall SEC-QOL scores (worse overall quality of life) revealed several statistically significant variables. A “not good” relationship with their husbands, “heavy” menstrual flow, and premenstrual breast symptoms (breast pain, hypersensitivity, and increased size) were identified as risk factors for poor quality of life due to their association with the highest scores (poor quality of life). The Arabic version of the modified SEC-QOL is a valid and reliable tool to measure the quality of life of Saudi contraceptive users.

## 1. Introduction

Contraception is defined as any intervention that reduces the chances of pregnancy after sexual intercourse [1]. There are several types of contraceptives, such as pills, male condoms, female condoms, hormonal intrauterine devices (IUDs), copper IUDs, periodic abstinence, implants, injections, and the lactation amenorrhea method. Assessing the quality of life (QoL) of contraceptive users is important to help them select appropriate contraceptive methods. In 2018, 32.9% of married Saudi women aged 15–49 years used contraceptives [2], with pills and IUDs (hormonal and copper) being the most common methods [3]. The main determinants for using contraceptives are age, marital status, educational level, monthly family income, and parity [4,5]. Saudi women choose contraception based on medical practitioners’ advice, family and friends’ subjective experiences, efficacy, and fear of adverse effects or complications [6]. Such complications or adverse effects are not limited to clinical symptoms but affect the subjective experience, including QoL. Therefore, it is important to assess the QoL among Saudi contraceptive users because it reflects their choice of contraceptives and perceptions of their physical, functional, and mental health status and the nonmedical aspects of their lives [7].

Contraceptives affect the QoL of contraceptive users in various aspects, including physiological, sexual, psychological, and social activity. The Spanish Society of Contraception Quality of Life (SEC-QOL) questionnaire was selected from other available women’s health QoL questionnaires because it comprehensively evaluates QoL across physiological, sexual, psychological, social activity, and breast-related domains and is the only available instrument that captures these multifaceted aspects of QoL. Spanish researchers designed the SEC-QOL questionnaire to assess the effect of contraceptive methods on health-related QoL of women. In 2011, the original version was published in Spanish and English and is widely used in Spain [8]. The SEC-QOL was used in a Spanish study (2016) to evaluate the QoL of women using a 52 mg levonorgestrel intrauterine system for contraception [9], in another Spanish study to assess university students’ health-related QoL and contraception use [10], and in a 2022 Jordanian study to assess the QoL of Jordanian women [11].

For the population-based aspects of this study, the comprehensive QoL assessment using the SEC-QOL questionnaire is crucial. The insights gained can inform authorities as they develop QoL programs as part of the broader 2030 vision for the community [12]. To our knowledge, only one study in Saudi Arabia compares the QoL of women using IUDs and pills using the WHO’s QOL-Bref (WHOQOL-BREF) scale [13]. However, no studies in Saudi Arabia used the SEC-QOL to identify the risks for poor QoL, and there is no Arabic version that is culturally appropriate [12]. Therefore, this study aimed to assess and identify the risks for poor QoL among Saudi contraceptive users.

## 2. Materials and Methods

This descriptive cross-sectional questionnaire-based study was conducted in Saudi Arabia between August and December 2023. The target participants were Saudi women from the general population (central, southern, eastern, western, and northern regions). They were identified through assigned data collectors in each area. The eligible participants for this study were randomly selected from the general population across various regions of Saudi Arabia. Data collectors in each geographic area were tasked with identifying and inviting these individuals to participate in the study. To facilitate this process, data collectors distributed an online self-administered questionnaire to the randomly selected participants by providing them with a barcode or link. Participants were thoroughly informed about the study’s objectives and were invited to partake in the research through this online questionnaire distribution method. Data were exported to a Microsoft Excel spreadsheet file using Google Docs to process and analyze the information.

### 2.1. Sample Size and Sampling Technique

Using OpenEpi^®^ version 3.0 (Centers for Disease Control and Prevention [CDC], Atlanta, GA, USA), the required sample size was 385, using the single proportion formula with a 5% margin of error, a confidence interval (CI) of 95%, and a population of 34 million (Saudi population) [12]; assuming the non-response rate is 10%, the sample size is equal to ~400. A *p*-value < 0.05 was considered statistically significant. A non-probability convenience sampling technique was employed.

### 2.2. Inclusion Criteria and Exclusion Criteria

The eligibility criteria included Saudi women aged 18 years and older who consented to participate. This encompassed woman of reproductive age who were currently using contraceptives, as well as those who had used them in the past, regardless of their menopausal status. Non-Saudi women, men, those aged <18 years, incomplete questionnaires, those who did not use contraceptives, and non-Arabic speakers were excluded.

### 2.3. Data Collection Instrument and Process

A cultural modification was made to the English version of the SEC-QOL [8]. We modified some statements for better understanding of the questions (Sex2, Sex4, Psy5, Soc1, Soc3) and added statements to the physiological domain (Ph4, Ph5) about weight changes and craving sweets (Appendix A). The questionnaire has three sections. Sociodemographic characteristics (age, educational level, monthly income, employment, marital status, description of relationship, parity, and abortions), menstruation characteristics such as contraception methods and premenstrual and menstrual changes, and a last section containing 20 items from the SEC-QOL questionnaire to assess the QoL of the participants using five subdomains: physiological symptoms (five items), sexual symptoms (four items), psychological symptoms (five items), social and daily activities (four items), and breast symptoms (two items). This questionnaire used a four-point Likert scale; the responses were scored as “Always” = 4, “Usually” = 3, “Sometimes” = 2, and “Never” = 1.

The cultural modification process involved a comprehensive review of the SEC-QOL instrument to identify and address any cultural biases or inappropriate content, professional translation of the questionnaire into Arabic with back-translation to verify conceptual equivalence, cognitive interviews with Saudi women to assess comprehension and relevance of the adapted items, and a pilot study with 20 Saudi women to evaluate the psychometric properties and feasibility of the revised questionnaire prior to full-scale implementation. The online survey included a short introductory message describing the aims of the study and obtained participants’ written consent prior to their participation. Questionnaires were completed anonymously for confidentiality, and participants could withdraw at any time. The Institutional Review Board Committee of King Saud University granted ethical approval before initiating the study (reference No. E-23-7865).

### 2.4. Statistical Analysis

Data analysis was conducted using RStudio software (RStudio version 4.3.0, Boston, MA, USA). Categorical data are presented as frequencies and percentages, and continuous variables are expressed as medians and interquartile ranges. The validity of the Arabic version of the modified SEC-QOL questionnaire was assessed by incorporating relevant items into an exploratory factor analysis and a subsequent confirmatory factor analysis. Internal consistency was assessed using Cronbach’s alpha coefficients for the overall questionnaire and its five subdomains. The subdomains were further converted to their respective scores by calculating the average responses of the coded items (Always = 4 to Never = 1). Therefore, the average score for each statement of the subdomain ranged from 1 to 4. The risk factors for each subdomain and the overall questionnaire were assessed by fitting generalized linear models using scores of the overall SEC-QOL and its subdomains (each in a separate model). The demographic and menstrual characteristics of the participants were included as independent variables, selected based on the backward selection method to best fit the models. The final independent variables are shown in the respective regression tables. The regression analysis results are expressed as beta coefficients and 95% CIs. Statistical significance was *p* < 0.05.

## 3. Results

### 3.1. Demographic Characteristics

We distributed 1077 online questionnaires, and 652 questionnaires were returned; 425 were excluded (424 due to ineligibility; 1 was incomplete). Therefore, the study population was 652 participants, and 44.5% of them were 31–40 years old. A large proportion of participants held a diploma or bachelor’s degree (75.5%), had a monthly income of <5000 Saudi riyal (SAR) (40.3%), were employed (45.4%), and were married and lived with their husbands (90.6%). Respondents described their relationships with their husbands as “good” (50.2%) and “somewhat good” (41.7%). The median number of children was 3.0. Most participants had not had any abortions (68.8%) or stillbirths (86.5%) (Table 1).

### 3.2. Menstruation Characteristics

The most common contraception used was the oral pill (51.5%). Participants reported a “normal” amount of blood during menstruation (75.5%) and used a “regular flow” pad (63.5%), and 53.8% of respondents reported blood clots during menstruation. Most participants reported regular menstruation (80.2%). Respondents reported breast-related premenstrual symptoms such as pain (52.8%), hypersensitivity (51.8%), and increased size (39.7%) (Table 1).

### 3.3. Factor Analysis

We collected data from participants on 20 items from the Spanish questionnaire (Appendix A). An exploratory factor analysis (EFA) was conducted on these 20 items using the varimax rotation method. Seven items were excluded because their factor loadings were statistically insignificant (Ph4, Ph5, Sex1, Sex2, Psy1, Psy2, Psy4). The final EFA model comprised 13 items (Table 2) that significantly loaded onto five subdomains (factors). A confirmatory factor analysis was conducted and the data had adequate fitting criteria (χ^2^ = 116.78, degrees of freedom = 55, comparative fit index = 0.981, Tucker–Lewis index = 0.973, root mean square error of approximation = 0.042, standardized root mean square = 0.033, *p* < 0.0001).

### 3.4. Reliability

Analysis of the questionnaire’s internal consistency revealed that the overall reliability (>0.7) was adequate (Cronbach’s alpha = 0.845, *n* = 13 items). This was also applicable to the questionnaire subdomains, including physiological symptoms (Cronbach’s alpha = 0.708, *n* = 3), psychological symptoms (Cronbach’s alpha = 0.913, *n* = 2), social symptoms (Cronbach’s alpha = 0.806, *n* = 4), and breast symptoms (Cronbach’s alpha = 0.744, *n* = 2). However, the sexual symptoms subdomain had a Cronbach’s alpha of 0.479.

### 3.5. Risk Factors for Higher Overall Spanish Society of Contraception Quality of Life Scores

Analysis of risk factors associated with higher overall SEC-QOL scores (indicating worse overall QoL) revealed several variables with statistical significance (*p* < 0.05). Participants with a “not good” relationship with their husbands had a notably higher SEC-QOL score (beta = 0.44, 95% CI: 0.27 to 0.62, *p* < 0.001), while those with a “somewhat good” relationship also showed a significant increase in the SEC-QOL score but lower scores than those who reported a “not good” relationship (beta = 0.18, 95% CI: 0.09 to 0.27, *p* < 0.001). Women with a “heavy” menstrual flow had high SEC-QOL scores (beta = 0.30, 95% CI: 0.15 to 0.45, *p* < 0.001). Participants with breast pain as a premenstrual symptom had significant increases in their SEC-QOL scores (beta = 0.33, 95% CI: 0.24 to 0.42, *p* < 0.001), as did those who reported hypersensitivity (beta = 0.10, 95% CI: 0.01 to 0.19, *p* = 0.029) and increased breast size (beta = 0.21, 95% CI: 0.12 to 0.30, *p* < 0.001). Individuals with irregular menstruation showed decreased SEC-QOL scores (beta = −0.11, 95% CI: −0.22 to 0.00, *p* = 0.043) (Table 3).

### 3.6. Risk Factors for Higher Physiological Symptom Scores

Among menstrual characteristics, participants with a “heavy” menstrual flow were significantly associated with higher physiological symptom scores (beta = 0.40, 95% CI: 0.19 to 0.62, *p* < 0.001). “Periodic abstinence” (beta = 0.29, 95% CI: 0.07 to 0.50, *p* = 0.009), injections (beta = 0.79, 95% CI: 0.30 to 1.28, *p* = 0.002), and hormonal IUDs (beta = 0.32, 95% CI: 0.02 to 0.63, *p* = 0.040) were associated with higher physiological symptom scores. Participants with a “somewhat good” relationship with their husbands exhibited higher physiological symptom scores (beta = 0.19, 95% CI: 0.06 to 0.31, *p* = 0.005) compared to those with a “good” relationship (Table 3).

### 3.7. Risk Factors for Higher Sexual Symptom Scores

Participants with a “not good” relationship with their husbands had significantly higher sexual symptom scores (beta = 0.52, 95% CI: 0.26 to 0.78, *p* < 0.001), and those with a “somewhat good” relationship also displayed an increase in sexual symptom scores (beta = 0.25, 95% CI: 0.12 to 0.39, *p* < 0.001) compared to those with a “good” relationship. Among contraception methods, using copper IUDs was associated with higher sexual symptom scores (beta = 0.33, 95% CI: 0.11 to 0.54, *p* = 0.003), and using male condoms exhibited a significant decrease in sexual symptom scores (beta = −0.19, 95% CI: −0.38 to −0.01, *p* = 0.044). Participants with pain during menstruation had higher sexual symptom scores (beta = 0.16, 95% CI: 0.04 to 0.29, *p* = 0.013) (Table 3).

### 3.8. Risk Factors for Higher Psychological Symptoms Scores

Participants with a “not good” relationship with their husbands had significantly higher psychological symptom scores (beta = 0.78, 95% CI: 0.47 to 1.10, *p* < 0.001), as did those with a “somewhat good” relationship (beta = 0.26, 95% CI: 0.10 to 0.42, *p* = 0.001) compared to those with a “good” relationship. A heavy menstrual flow was associated with higher psychological symptom scores (beta = 0.39, 95% CI: 0.06 to 0.72, *p* = 0.020) compared to a light flow (Table 3).

### 3.9. Risk Factors for Higher Breast Symptom Scores

Currently employed participants had significantly higher breast symptom scores (beta = 0.19, 95% CI: 0.05 to 0.32, *p* = 0.007) compared to the currently unemployed. Participants who had experienced 1–2 abortions had lower breast symptom scores (beta = −0.18, 95% CI: −0.35 to −0.02, *p* = 0.031) than those with no abortions. Participants with 3–4 abortions had higher scores than participants with ≤2 abortions in breast symptom scores but the difference was not statically significant (*p* = 0.790). In menstrual characteristics, having a “heavy” menstrual flow was associated with higher breast symptom scores (beta = 0.37, 95% CI: 0.14 to 0.60, *p* = 0.002) compared to those who had light bleeding (Table 3).

## 4. Discussion

Studies in Europe, Japan, Spain, and Jordan have used the SEC-QOL to assess the QoL among contraceptive users [8,9,10,11]. This study aimed to determine the reliability of the Arabic version of the modified SEC-QOL among the Saudi population. Our results are consistent with a study in Jordan using the same scale [11]. We found that the Arabic version had a Cronbach’s alpha coefficient of 0.845, indicating it as a reliable tool for QoL measurement of Saudi contraceptive users. However, our results show that the Cronbach’s alpha coefficient was the lowest for sexual symptoms.

Our results show that participants who did not have good relationships with their spouses, had heavy menstrual flow, and had premenstrual symptoms (breast pain, hypersensitivity, and increased breast size) had higher scores on the SEC-QOL (worse QoL). Those with regular menstruation scored lower on the SEC-QOL (better QoL); these differences were statistically significant. Earlier studies suggest that the use of hormonal contraceptives mitigate these risk factors by regulating the menstrual cycle and positively influencing QoL [10,14,15]. However, the current study did not directly examine the relationship between hormonal contraceptive use and QoL outcomes. Further research is needed to fully understand interventions, such as hormonal contraceptives, and may help improve the QoL for individuals experiencing menstruation-related symptoms and relationship challenges.

Our findings regarding the risk factors for higher overall SEC-QOL scores revealed that women using non-hormonal contraceptives reported higher scores for physiological menstrual symptoms compared to those using hormonal contraceptives. These results align with previous studies [10,14,15]. However, one clinical trial demonstrated that ethinyl estradiol contraceptive pills significantly reduced physiological symptoms, such as dysmenorrhea during menstruation, and helped regulate the menstrual cycle, ultimately having a positive impact on women’s quality of life [16].

Women using copper IUDs had more sexual symptoms than those whose partners used male condoms. A systematic review reported females with copper IUDs can experience increased blood flow, unscheduled bleeding, and cramping with the device, which may affect their sexual pleasure [17]. Our study indicated that participants with pain before menstruation scored higher (poorer QoL) on measures in the sexual domain compared to those with regular menstrual cycles. This finding is consistent with research showing that patients with an earlier onset of dysmenorrhea (<12 years old) tend to have severe menstrual pain, and there were significant correlations between the severity of menstrual pain and sexual activity [18].

Participants without a good relationship with their husbands had significantly higher psychological symptom scores than participants with a good relationship, which is consistent with previous studies on Saudi working married women. They reported that those experiencing marital dissatisfaction had higher levels of emotional distress and women married for a longer duration and higher parity were more likely to report higher levels of “emotional divorce”—a psychological distancing from the marital relationship [19]. This aligns with findings that increased marital duration and number of children can contribute to greater marital strain and dissatisfaction over time, potentially leading to a form of emotional detachment between spouses [19]. Taken together, these findings underscore the pivotal role of the marital relationship in shaping the psychological health of married women and highlight the need for interventions to support relationship quality, especially for long-term married couples with children. Our findings that heavy menstrual flow is associated with higher psychological symptoms than a light menstrual flow are consistent with studies that reported that menstrual irregularity has a significant effect on psychological and social symptoms in Saudis [20,21].

Our findings that women with a heavy menstrual flow had higher scores for the breast domain compared to women with a light menstrual flow are consistent with findings that menstrual irregularities can cause mastalgia [22]. Our results reveal that women who had never had an abortion had higher scores for the breast domain compared to those who had had 1–2 abortions, and the difference is statistically significant. A study by Henkel et al. found that most women do not experience breast engorgement and tenderness after abortion, which is consistent with our findings [23].

The strength of this study is that it can be widely used to measure SEC-QOL, and the cultural/linguistic relevance of the study’s approach to measuring SEC-QOL among Arabic-speaking individuals enables more standardized and meaningful assessments within target populations, making it a valuable tool. It can help Saudi authorities with a transformation plan for health-related QoL programs. However, this study had some limitations. Like most questionnaire-based studies, there could be a reporting bias that influenced the responses from the participants. The questionnaire used for the study was relatively long, which could have led to fatigue and affected the quality of the responses. One of the study’s limitations is the low reliability observed in the sexual symptoms subdomain, which may be attributed to the lack of comprehensive educational programs on sexual health and marital counseling in Saudi Arabia.

## 5. Conclusions

Oral contraceptive pills are the most common contraceptive among Saudi women. The Arabic version of the modified SEC-QOL is a valid and reliable tool to measure the QoL of Saudi contraceptive users in Saudi Arabia. The risk factors for poor QoL among Saudi contraceptive users are the lack of good marital relationships, heavy menstrual flow, and premenstrual symptoms (breast pain, hypersensitivity, and increased breast size).

We recommend implementing a program to counsel and educate couples to support each other and improve their relationships, as it is the main risk factor for Saudi women using contraceptives. Further studies are needed to delve deeper into relationships between contraceptive use, premenstrual symptoms, and QoL. By addressing these gaps in knowledge, healthcare providers can better understand and address the specific risks and challenges of using contraception, leading to improved QoL outcomes.

## Figures and Tables

**Table 1 behavsci-14-00829-t001:** Participants’ characteristics.

Characteristic	N (%)
Age (years)	
<20	8 (1.2%)
21–30	237 (36.3%)
31–40	290 (44.5%)
41–50	104 (16.0%)
>50	13 (2.0%)
Education	
General education	90 (13.8%)
Diploma/bachelor’s degree	492 (75.5%)
Postgraduate	70 (10.7%)
Monthly income (SAR)	
<5000	263 (40.3%)
5000–10,000	167 (25.6%)
>10,000–20,000	153 (23.5%)
>20,000–30,000	41 (6.3%)
>30,000	28 (4.3%)
Currently working	
Yes	296 (45.4%)
No	356 (54.6%)
Living with husband	
Yes	591 (90.6%)
No	61(9.4%)
Relationship with husband	
Good	327 (50.2%)
Somewhat good	272 (41.7%)
Not good	53 (8.1%)
Number of children	3.0 (1.0–4.0)
Number of abortions *	
None	431 (68.8%)
1–2	129 (20.6%)
3–4	66 (10.5%)
Stillbirths	
Yes	88 (13.5%)
No	564 (86.5%)
Contraception used	
Pills	336 (51.5%)
Male condom	95 (14.6%)
Female condom	3 (0.5%)
Hormonal IUD	32 (4.9%)
Copper IUD	73 (11.2%)
Periodic abstinence	61 (9.4%)
Implants	38 (5.8%)
Injections	9 (1.4%)
Lactation amenorrhea method	5 (0.8%)
Blood flow during menstruation	
Very little	26 (4.0%)
Little	80 (12.3%)
Normal	492 (75.5%)
Severe bleeding	54 (8.3%)
Type of pad	
Light flow	145 (22.2%)
Regular flow	414 (63.5%)
Heavy flow	93 (14.3%)
Blood clots	
Yes	351 (53.8%)
No	301 (46.2%)
Regular menstruation	
Yes	462 (80.2%)
No	190 (19.8%)
Breast-related premenstrual symptoms	
Pain	344 (52.8%)
Hypersensitivity	338 (51.8%)
Increased size	259 (39.7%)

IUD = intrauterine device; * missing 26 records.

**Table 2 behavsci-14-00829-t002:** Factorial coefficients after rotation of the 13 items from the SEC-QOL questionnaire.

Variable	Factor 1	Factor 2	Factor 3	Factor 4	Factor 5
Ph_1	0.159	0.623	0.101	0.160	0.112
Ph_2	0.192	0.578	0.063	0.149	0.100
Ph_3	0.145	0.671	0.127	0.082	0.021
Sex_3	0.177	−0.003	0.126	0.000	0.564
Sex_4	0.068	0.163	0.052	0.087	0.527
Psy_3	0.246	0.160	0.868	0.073	0.147
Psy_5	0.330	0.171	0.809	0.102	0.145
Soc_1	0.717	0.363	0.215	0.084	0.152
Soc_2	0.674	0.406	0.204	0.140	0.149
Soc_3	0.506	0.026	0.139	0.025	0.073
Soc_4	0.628	0.199	0.149	0.162	0.155
Br_1	0.128	0.417	0.097	0.501	0.134
Br_2	0.149	0.210	0.084	0.960	0.042

Ph = physiological domain, Sex = sexual domain, Psy = psychological domain, Soc = social activity domain, Br = breast domain.

**Table 3 behavsci-14-00829-t003:** Risk factors for higher SEC-QOL scores.

Characteristic	Beta Coefficient	95% CI	*p*-Values
Risk factors for higher overall SEC-QOL scores			
Relationship with husband			
Good	Reference	Reference	
Somewhat good	0.18	0.09, 0.27	<0.001 *
Not good	0.44	0.27, 0.62	<0.001 *
Type of pad			
Light flow	Reference	Reference	
Regular flow	0.08	−0.03, 0.18	0.147
Heavy flow	0.30	0.15, 0.45	<0.001 *
Premenstrual breast symptoms			
Pain			
No	Reference	Reference	
Yes	0.33	0.24, 0.42	<0.001 *
Hypersensitivity			
No	Reference	Reference	
Yes	0.10	0.01, 0.19	0.029 *
Increased size			
No	Reference	Reference	
Yes	0.21	0.12, 0.30	<0.001 *
Regular menstruation			
No	Reference	Reference	
Yes	−0.11	−0.22, 0.00	0.043 *
*Physiological Domain*			
Relationship with husband			
Good	Reference	Reference	
Somewhat good	0.19	0.06, 0.31	0.005 *
Not good	0.08	−0.17, 0.33	0.507
Contraception used			
Pills	Reference	Reference	
Male condoms	0.11	−0.08, 0.29	0.253
Female condoms	−1.04	−2.47, 0.40	0.158
Hormonal IUDs	0.32	0.02, 0.63	0.040 *
Copper IUDs	0.19	−0.01, 0.40	0.065
Periodic abstinences	0.29	0.07, 0.50	0.009 *
Implants	0.04	−0.22, 0.31	0.738
Injections	0.79	0.30, 1.28	0.002 *
Lactation amenorrhea method	−0.10	−0.92, 0.73	0.819
Type of pad			
Light flow	Reference	Reference	
Regular flow	0.17	0.02, 0.32	0.026 *
Heavy flow	0.40	0.19, 0.62	<0.001 *
*Sexual Domain*			
Relationship with husband			
Good	Reference	Reference	
Somewhat good	0.25	0.12, 0.39	<0.001 *
Not good	0.52	0.26, 0.78	<0.001 *
Contraception used			
Pills	Reference	Reference	
Male condom	−0.19	−0.38, −0.01	0.044 *
Female condom	−0.06	−1.54, 1.43	0.941
Hormonal IUD	−0.01	−0.33, 0.32	0.966
Copper IUD	0.33	0.11, 0.54	0.003 *
Periodic abstinence	−0.16	−0.38, 0.06	0.162
Implants	0.11	−0.18, 0.40	0.461
Injections	0.54	0.03, 1.05	0.040 *
Lactation amenorrhea method	0.55	−0.31, 1.41	0.214
Pain			
No	Reference	Reference	
Yes	0.16	0.04, 0.29	0.013 *
*Psychological Domain*			
Relationship with husband			
Good	Reference	Reference	
Somewhat good	0.26	0.10, 0.42	0.001 *
Not good	0.78	0.47, 1.10	<0.001 *
Menstruation amount			
Light amount	Reference	Reference	
Regular amount	0.10	−0.11, 0.32	0.351
Heavy amount	0.39	0.06, 0.72	0.020 *
*Breast Domain*			
Currently working			
No	Reference	Reference	
Yes	0.19	0.05, 0.32	0.007 *
Number of abortions			
None	Reference	Reference	
1 to 2	−0.18	−0.35, −0.02	0.031 *
3 to 4	0.03	−0.19, 0.26	0.790
Type of pad			
Light flow	Reference	Reference	
Regular flow	0.10	−0.06, 0.26	0.225
Heavy flow	0.37	0.14, 0.60	0.002 *

CI = confidence interval, IUD = intrauterine device.; * *p*-values (<0.005).

## Data Availability

The data that support the findings of this study are available from the corresponding author, L.R.B., upon request.

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
