# Peer review of "Risk Factors for Low Quality of Life among Women Using Different Types of Contraceptives in Saudi Arabia: A Questionnaire-Based Study"

_behavsci, 2024, doi:10.3390/bs14090829_

Round 1
Reviewer 1 Report
Comments and Suggestions for Authors
Do the authors have some explanation, for why the sexual symptoms subdomain has low reliability? If yes, please mention it.
The visual appearance of Figure 1 can be improved.
Author Response
Dear reviewer,
We sincerely thank you for taking the time to review our manuscript and providing constructive feedback to improve our manuscript. We have revised the manuscript accordingly by following the reviewers’ suggestion and changes in the manuscript were highlighted with yellow.
Reviewer 1 comments:
1-Do the authors have some explanation, for why the sexual symptoms subdomain has low reliability? If yes, please mention it.
Response 1: Thank you for your comment. We acknowledged the low reliability in the sexual symptom’s subdomain as a limitation of the study (page no. 10, lines 292-295). We suggest that the lack of comprehensive educational programs on sexual health and marital counseling in the Saudi Arabian context may be a contributing factor. The conclusion section of the manuscript recommends the development and implementation of such educational initiatives to better support couples and improve the assessment of sexual well-being.
2- The visual appearance of Figure 1 can be improved.
Response 2: Thank you for your comment. After careful consideration, we have decided to remove this figure from the manuscript. The rationale behind this decision is that the quality scores are comprehensively detailed in Table 2, along with a thorough explanation provided in the results section on page no.5, lines 169-174.
We believe that the information presented in Table 2 sufficiently conveys the necessary insights regarding the quality scores, making the additional boxplot redundant. This adjustment aims to enhance the clarity and conciseness of our manuscript without compromising the quality of the information provided.
Reviewer 2 Report
Comments and Suggestions for Authors
In this study the authors tested the validity and reliability of the SEC-QOL as a tool to assess quality of life among contraceptive users in Saudi Arabia, and examined the related possibly influencing factors. The authors examined data collected through an online questionnaire from over 650 women. The study findings support the reliability and validity of the scale in this population, and found sociodemographic and menstrual-related factors as influencing the quality of life of these women. The paper is well-written, and the findings are of public health interest; some comments to improve and clarify the meaning of the manuscript are reported below:
1. Material and Methods, page 2, lines 72-73, “They were identified through assigned data collectors in each area. The data collectors sent an online self-administered questionnaire using a barcode…”. Please, provide more details on how the eligible participants were selected. Where were they picked from? Were they randomly picked from a community, University, general population, GP clients, or what else? How were they informed about and invited to participate?
2. Material and Methods, page 2, lines 83-86: didn’t the authors apply any upper-age limit? As it is stated now, it appears that any women who had used contraceptives at any lifetime point could participate (in other words, also older menopausal women who had used contraceptives many years before, which was likely not the case). What does “who had previously used contraceptives” mean? I think a specific timeframe for the use of contraceptives (and possibly for the women’s age) was in fact settled. Please, clarify.
3. Discussion, page 9, lines 241-245: the interpretation of this sentence is not clear. Where are these results reported? Also, if EE pills reduce physiological symptoms, this is in line with the study findings of non-hormonal users having higher scores (worse QoL) on physiological symptoms; as the sentence is written now, it appears that the study findings are opposite to those of reference [16]. Please, clarify.
4. In general, the focus of the study is on the validation of the SEC-QOL in the Saudi population, and, apparently as a secondary aim, the examination of factors related to QOL in this population. Although the SEC-QOL is designed specifically for women using contraceptives, the study aim is not on the specific relationship between contraception use and quality of life. Maybe make this clear in the title? E.g., “Risk factors for low quality of life among women using different types of contraceptive in Saudia Arabia: a questionnaire-based study”, or something similar.
Minor:
1. Paragraph 2.4: perhaps the correct heading in “Statistical analysis” rather than “Statistical sample size”.
2. A legend explaining Figure 1 could help the reading.
Comments on the Quality of English LanguageThe paper could benefit from a minor language check. E.g., Please, check the language in the sentence at lines 232-234 (starting with “While”)
Author Response
Dear Reviewer,
We sincerely thank you for taking the time to review our manuscript and providing constructive feedback to improve our manuscript. We have revised the manuscript accordingly by following the reviewers’ suggestion and changes in the manuscript were highlighted with yellow.
1- Material and Methods, page 2, lines 72-73, “They were identified through assigned data collectors in each area. The data collectors sent an online self-administered questionnaire using a barcode…”. Please, provide more details on how the eligible participants were selected. Where were they picked from? Were they randomly picked from a community, University, general population, GP clients, or what else? How were they informed about and invited to participate?
Response 1: Thank you for your valuable comments. In response, we would like to clarify that the eligible participants were randomly selected from the general population across various regions of Saudi Arabia. Data collectors in each geographic area were tasked with identifying and inviting participants to take part in the study.
The data collectors distributed an online, self-administered questionnaire to the randomly selected individuals using either a barcode or a link specific to their respective areas. Participants were informed about the study and invited to participate through this online distribution method.
As requested, we have updated the methodology section to reflect these details. Please refer to page 2, lines 73-80.
2- Material and Methods, page 2, lines 83-86: didn’t the authors apply any upper-age limit? As it is stated now, it appears that any women who had used contraceptives at any lifetime point could participate (in other words, also older menopausal women who had used contraceptives many years before, which was likely not the case). What does “who had previously used contraceptives” mean? I think a specific timeframe for the use of contraceptives (and possibly for the women’s age) was in fact settled. Please, clarify.
Response 2: Thank you for your insightful observation regarding the absence of an upper-age limit for participants in the manuscript. You correctly noted that the inclusion criteria specified "women who had previously used contraceptives" without delineating a specific timeframe for contraceptive use. This lack of specification could indeed have encompassed menopausal women who had used contraceptives many years prior. We appreciate your attention to this detail, as it is a crucial consideration for the study’s design.
The decision not to impose an upper-age limit was intentional, aimed at ensuring a comprehensive representation of the target population. This included both women of reproductive age currently using contraceptives and those who had used them in the past, irrespective of their menopausal status. We believed that excluding menopausal women who had previously used contraceptives might introduce bias into the sample and restrict the generalizability of the findings. However, we recognize that the manuscript's wording may have been unclear on this point, and we will revise it to provide greater clarity (page no.2, lines 91-93).
3- Discussion, page 9, lines 241-245: the interpretation of this sentence is not clear. Where are these results reported? Also, if EE pills reduce physiological symptoms, this is in line with the study findings of non-hormonal users having higher scores (worse QoL) on physiological symptoms; as the sentence is written now, it appears that the study findings are opposite to those of reference [16]. Please, clarify.
Response 3: The sentence was referring to the results related to "risk factors for higher overall SEC-QOL scores" reported in the manuscript. As you mentioned, the relevant study cited in reference [16] explained that the use of hormonal contraceptive pills, such as ethinyl estradiol (EE) pills, can regulate menstruation and have a positive impact on women's quality of life. So, the hormonal contraceptive will have better QoL than non-hormonal users.
The paragraph was revised and rewritten on page no.9, lines 247-253.
4-In general, the focus of the study is on the validation of the SEC-QOL in the Saudi population, and, apparently as a secondary aim, the examination of factors related to QOL in this population. Although the SEC-QOL is designed specifically for women using contraceptives, the study aim is not on the specific relationship between contraception use and quality of life. Maybe make this clear in the title? E.g., “Risk factors for low quality of life among women using different types of contraceptives in Saudi Arabia: a questionnaire-based study”, or something similar.
Response 4: Thank you for your comment. The suggested title was used.
5- Paragraph 2.4: perhaps the correct heading in “Statistical analysis” rather than “Statistical sample size”.
Response 5: Thank you for your comment. It was corrected as requested (page no.3, lines 119). Als, an editing certificate was attached for your convenience.
6- A legend explaining Figure 1 could help the reading.
Response 6: We appreciate your feedback regarding the inclusion of the legend of figure 1 depicting the boxplot of the quality scores. After careful consideration, we have decided to remove this figure from the manuscript. The rationale behind this decision is that the quality scores are comprehensively detailed in Table 2, along with a thorough explanation provided in the results section on no.5, lines 169-174.
We believe that the information presented in Table 2 sufficiently conveys the necessary insights regarding the quality scores, making the additional boxplot redundant. This adjustment aims to enhance the clarity and conciseness of our manuscript without compromising the quality of the information provided.
